# Friendly Residential Environments That Generate Autonomy in Older Persons

**DOI:** 10.3390/ijerph20010409

**Published:** 2022-12-27

**Authors:** Alejandra Segura Cardona, Doris Cardona Arango, Angela Segura Cardona, Carlos Robledo Marín, Diana Muñoz Rodríguez

**Affiliations:** 1Faculty of Psychology, Universidad CES, Medellín 050001, Colombia; 2Faculty of Medicine, Universidad CES, Medellín 050001, Colombia; 3Graduate School, Universidad CES, Medellín 050001, Colombia; 4Fundación Opción Colombia (FUNDACOL), Medellín 050001, Colombia

**Keywords:** aged, self-sufficiency, motor skills, social environments

## Abstract

Objective: This study aimed to explore the housing and residential environment conditions associated with functional autonomy in older persons. Methods: A quantitative cross-sectional study was conducted, including 175 individuals over the age of 60. Participants were non-institutionalized urban residents of Medellín, Colombia, selected by random two-stage sampling (neighborhoods and blocks). Analysis was done according to functional autonomy of action (dependent variable); and demographic conditions, housing, and the physical and social environment suggested by the World Health Organization (WHO) in the strategy of age-friendly cities (independent variables). Univariate, bivariate, and multivariate analyses were performed with these variables, where the odds ratio (OR), association hypothesis test, and confidence intervals were estimated, using logistic regression models. Results: 89.7% of older persons had moderate physical performance. The performance of intergenerational activities (OR = 5.28) and community actions (OR = 11.28) were part of social environments. The adaptations in public transport (OR = 90.33), sanitary services (OR = 4.1), and lighting in parks (OR = 19.9) of the physical environment were the associations found with functional autonomy. Conclusions: Exploring how the physical and social environments surrounding housing are associated with the functional performance of older persons can generate useful information to support public health and city infrastructure strategies that improve their physical performance and maintain autonomy.

## 1. Introduction

In the Decade of Healthy Aging 2021–2030, a new concept emerges, based on the necessary conditions to act autonomously and independently in old age, adding quality to quantity of life, allowing older persons to develop social, functional, and interpersonal skills in the environments that surround them [1,2]. Better individual (physical, mental, and functional) and environmental (family and social) conditions allow the enjoyment of old age with greater well-being.

To achieve this, the World Health Organization (WHO) has considered the design of age-friendly cities that promote healthy lifestyles; social participation, entertainment, volunteering, and employment has been considered, as well as accessibility to buildings, public spaces, public transport, friendly residences (residential environments with safe and comfortable infrastructures for the aging population), and social services friendly to older persons (health, participation, and mobility, among others) [3].

The implementation of age-friendly environments is a priority, considering the increase in the number of older persons in the world, going from 11% to 22% between 2000 and 2050 [4]. Likewise, in Colombia, the population of 60 years and over has seen a constant increase; in 2018, it represented 13.2% of the total population, and according to estimates, by the year 2050, it will be around 20%. The city of Medellín has a higher percentage of older persons [5]; however, in its planning, the demographic transformations [6] and the increase in the vehicle fleet have not been considered, nor has the adequacy of parks, public squares, streets, sidewalks, or homes.

It is a challenge for society to recognize its commitment to older people and provide them with age-friendly environments with better physical and social conditions. This promotes an old age with dignity and the autonomy to be, have, and do what makes sense to the older person. The founding spirit of the Decade of Healthy Aging seeks to maintain physical and mental function (intrinsic and extrinsic functional capacity) [7], and a commitment to a prosperous old age (well-being, happiness, autonomy, and quality of life) [8].

In Colombia, this is embodied in the National Public Policy on Aging and Old Age 2022–2031 [9], which is protected by the Inter-American Convention on the Protection of the Human Rights of Older Persons [10]. It promotes healthy aging to achieve an independent, autonomous, and productive life in old age. This is achieved through the strengthening of social determinants, health, and environmental settings in the community, home, educational, work, and institutional settings [9].

Residential environments have vital importance for older people [11], as they are fundamental for the promotion of their autonomy in two ways. First, autonomy is understood as the capacity and freedom for decision-making, without external coercion [12]. Second, functional autonomy is seen as the autonomy of action (physical independence), the autonomy of the will (self-determination), and the autonomy of thought (the ability to judge any situation) [13]. It is in this second way that this research is based, since it recognizes autonomy of action and physical independence for the realization of activities, mediated by its ability to achieve them.

Functional capacity is based on physical performance and physical parameters (balance, walking speed, and ability to switch from sitting to bipedal) and although its maintenance facilitates the development of individual skills, it cannot be ignored that individual health outcomes are conditioned by physical and social environments [14,15,16]. Therefore, in this study, physical performance [17] was measured with the short battery of physical performance (SPPB): mobility, balance, muscle strength, and dexterity, as they were considered predictors of functional capacity [18] and mortality [19] in older persons. This is under the premise that it is in the residential environment where autonomy, physical independence, and functional capacity in old age are most favored [20].

This study sought to explore the housing and residential environment conditions associated with the functional autonomy of older persons.

## 2. Materials and Methods

Study design. A quantitative cross-sectional study [21] was conducted in which 175 people over the age of 60 and their residential environments were surveyed. Participants had to be non-institutionalized, their homes located in the urban area of the city of Medellín Colombia, and they had to agree to participate in the study.

Sampling. The city of Medellín (Colombia) is the capital of the Department of Antioquia, and is divided into 16 communes or conglomerates and 264 neighborhoods. With cartographic information, a probabilistic sampling was designed, through random clusters in two stages. The selection of neighborhoods was made within each of the 16 communes, with random systematic sampling. Three neighborhoods per commune were selected for a total of 51 neighborhoods, as secondary sampling units (SSU). Within each neighborhood, three blocks were selected as primary sampling units (PSU) by simple random sampling, for a total of 175 blocks. Three older individuals residing in the block were randomly taken, as the final sampling unit (FSU), and their residential environments (physical and social) were studied. Sample size was calculated with a 95% confidence and a precision for results of 3.6%. The percentage of non-response was 48.2%, given the time of global health emergency that prevented access to nursing homes for older persons.

Information gathering. The information was collected between the months of April and August 2021. A form that included demographic variables and residential environments, collected with three instruments, was used. They included individual conditions; functional autonomy of action taken with the short physical performance battery (SPPB) [22]; perception of the physical and social environments, using the WHO age-friendly cities guide (information from participants); and built environment, through systematic social observation (SSO), which allowed recording actions, activities, and behaviors in areas of social agglomeration (parks, neighborhoods, etc.) [23].

Dependent variable: Functional autonomy (SPPB). Autonomy was measured through the validated version for Colombia of the Short Physical Performance Battery (SPPB) [24] which is one of the most widely used instruments to measure physical performance in population-based aging studies. It consists of three tests: hierarchical balance assessments lasting ten seconds, short walking (4 meters at a usual pace), and the ability to get up from a chair five consecutive times. SPPB can be safely used to assess functional capacity in clinical and outpatient settings. Each test ranges from 0 to 4, and the final score ranges from 0 to 12. A higher score means better functional capacity. For this study, and according to the use of the National Institute on Aging, those scores below 6 were classified as “low and very low physical performance”, while moderate performance was between 7 to 9 and high performance (satisfactory) between 10 and 12 [25].

Independent variables. Among them, some that required the use of validated scales were included. Family functioning was measured through the APGAR familiar score [26], to classify participants into those with good family functioning and those with dysfunction. Likewise, social support was measured through the Medical Outcomes Study (MOS) questionnaire [27], to determine whether there is poor support. The social, physical, and built environments were measured as perceived by the older person in a Yes or No question.

Statistical analysis. Univariate, bivariate, and multivariate analyses were performed. A univariate analysis was used in the characterization of older persons, and a description of their residential environment was carried out. Proportions were used in the estimation of physical performance (Short Physical Performance Battery (SPPB)) as a proxy variable of functional autonomy, comparing moderate and low/very low performances. For the bivariate analysis, non-causal associations were made between functional autonomy and demographic and environmental variables, for which the hypothesis test (association Chi-square, X^2^) was used with a level of statistical significance of 5% and odds ratio (OR) [28] with a 95% confidence interval (95% CI), performed with binary logistic regression models. Finally, a multivariate analysis adjusted for sex, age, marital status, and type of housing was performed and the coefficient of determination (R^2^) was calculated.

Ethical considerations. This project was part of an overall project entitled “Salud y bienestar mental de la persona mayor, en cinco ciudades de Colombia. Año 2020”, funded by the Ministry of Science and Technology and CES University, and was approved by the Ethics Committee of the same institution. The principles established in the Declaration of Helsinki [29], Resolution 008430 of 1993 [30], and Decree 1581 of 2012 [31] were preserved.

## 3. Results

### 3.1. Individual Characteristics of Older Persons

Of the 175 older persons surveyed, there was a predominance of women (64%), aged between 60 and 70 (68.6%), and two out of three were single (single, separated, divorced, or widowed) at the time of inclusion in the study. Regarding their family environment, they declared that they were living in a functional home (64.6%), with 10.3% reported having been rejected by a health institution in the past (Table 1). 

Functional autonomy of action, according to the SPPB scale, was found to be moderate in 89.7% (157), similar to what has been previously described: mainly women, under 70 years of age, without a stable partner, homeowners, in functional homes, without perceiving rejection from health institutions, with no statistically significant difference (Table 1).

### 3.2. Perception of the Social Environment (WHO Age-Friendly Cities Guide)

Participants reported perceiving that their neighborhoods are not friendly to older residents. They have negative opinions regarding the little enrolment in social and cultural events in the neighborhoods where they live (60%), and the schedules when and facilities where these events are offered (both aspects have close to 60% negative perception). Likewise, they feel a lack of opportunity to offer input and make decisions for the neighborhood through community associations (52.6%) or to participate in health campaigns (63.4%). More than half consider that encounters and social interactions between families and people of other ages are not promoted. There are no programs where older persons share their experience and knowledge with children and young people, and they feel their role in the city is not respected (60%) (Table 2).

In relation to the physical infrastructure of the spaces of coexistence and recreation, two out of three consider that parks and rest areas do not have adequate spaces for older persons to engage in leisure activities. Therefore, it is normal not to see many older persons in commercial and religious establishments. The vast majority (95.4%) perceive very few job opportunities for older persons and a lack of support by the community action boards (Juntas de Acción Comunal, JAC) for older persons who require assistance from long-stay centers, temporary homes, or those who require caregivers (Table 2).

Adequate housing with spacious and comfortable spaces for the mobility of older persons, whether with functional limitations or not, was well perceived (82.3%). Feeling safe to go out into the neighborhood (96.8%) can increase the probability of having functional autonomy. Although these findings are not statistically significant, conditions of residential environments friendly to older persons were observed. Likewise, it was found that city adaptations could favor autonomy (moderate or high). Spaces to share with people of other ages and having urban equipment in parks and common areas increase the probability of functional autonomy, and of being considered for health campaigns, mainly during the COVID-19 pandemic (Table 2).

### 3.3. Perception of the Physical Environment (WHO Age-Friendly Cities Guide)

The physical environment surrounding the house was also not well perceived by participants. They reported perceiving a great deal of visual pollution in nearby parks (66.9%) and noise pollution in common areas (54.9%). They also reported a lack of cleanliness, comfort, and airy public toilets for older persons and their companions, in case of they are needed, nor is there equipment (railings, non-slip sidewalks, chairs, lighting) in the sport or residential areas where they move (Table 3).

The access roads to housing were considered in more than 60% of perceptions to be lacking in sidewalks wide enough for free movement with canes, walkers, or wheelchairs, and in some cases, they are occupied by informal vendors, vehicles, and garbage, among others. In vehicular environments, participants reported that traffic lights have crossing times that are too short for an older person with a functional problem (69.1%) and they do not have adequate signage for all road users. The physical infrastructure that surrounds the houses and buildings in the neighborhood does not have enough ramps, elevators, and age-friendly rest areas, as perceived by around 70% of older persons. The city’s transportation services do not have differential rates (69.1%) and, in commercial places, priority is not given to older persons (71.4%) (Table 3).

In general, older persons’ perception is negative with respect to the urban design of their neighborhood and the city to achieve healthy behaviors, entertainment, accessibility to housing and common areas, which guarantee a dignified old age by having age-friendly residential environments. When this perception is positive, there is evidence of moderate functional autonomy of action, mainly when there is good access to public toilets, sports areas, buildings, parks, ramps, elevators, rest areas, and preferential rows and rates for public transport and commercial establishments, among others (Table 3). 

### 3.4. Perception of the Built Environment

The observation of the residential environment of the older person’s housing within the city was collected through the direct observation of streets, roads, sidewalks, greenery, graffiti, murals, gardens, museums, transport routes, lighting, and vehicular flow, among others. It showed a lack of separators between vehicular traffic and pedestrian traffic (84.6%), trees around the residential environment (65.1%), murals or gardens while walking or cycling (66.9%), and lamps or lampposts (90.3%) to improve visibility (Table 4). 

Regarding transit routes and the circulation of road users such as pedestrians, cyclists, motorcyclists, and vehicles, a high vehicular flow on the roads surrounding the residences (86.9%) was observed with several public transport routes, such as trains, Metrocable, Metrobus, trolleys, etc. (97.1%). Conditions for walking or cycling are not favorable (90.3%) due to the high volume (86.9%) and speed (94.9%) of vehicles, with no speed reduction devices (89.7%) or crossing aids, such as pedestrian-crossing traffic lights or crosswalk lines, and, therefore, no one was observed walking or cycling, during the observation tour (Table 4). 

The state of sidewalks was considered good in 35.4%, with the rest being under repair, having only one side in a working state, and in other cases, there were no sidewalks (42.3%). Only 40% of the sidewalks observed were continuous, allowing for better mobilization. The state of the roads shows many slopes; Medellín is a city built within the mountains with a broken topography, so it is normal to find slopes with a high inclination (23.4%), with the rest being moderate or slightly sloped. There was a predominance of a single lane (64.6%) in the block where the older person’s home is located, with up to five lanes (1.1%) in some cases (Table 4).

## 4. Discussion

The study found that older persons living in Medellín (Colombia) who participated had moderate physical performance and the perception of the factors of the physical and social environment in which they live was related to this outcome. There is enough evidence suggesting that self-reported perceptions of the neighborhood environment should be considered crucial components of active aging [16]. The relationship with physical performance is necessary because this involves activities, such as walking, sitting, and standing, that have a fundamental role in the ability of the older person to perform daily activities and stay socially involved, in addition to preventing the development of chronic conditions. Physical performance is recognized as a central element of successful aging [32].

The evidence of the relationship between the environment and outcomes such as physical activity, with achieving various goals in health and sustainable development worldwide, has strengthened [33]. It has been shown to be associated with outcomes such as falls, frailty, sarcopenia, and other chronic conditions [34], intrinsic factors of human health. Evidence is limited to factors specific to autonomy and the functional capacity of older persons where elements of the environment are involved. This is a pioneer study in Colombia: it explores how the physical and social environments, perceived and constructed, in which older persons reside may relate to physical performance considered as a functional measure of autonomy and, therefore, an achievement of active aging.

Although it was found that nine out of ten had moderate performance, none achieved high (satisfactory) performance, not exceeding nine points on the SPPB scale. Some studies in Latin America have documented low physical performance. In Peru, in 2019, a study found that four out of every ten older persons (44.5%) had low physical performance [35], but the factors they explored were of the individual order. As in in the findings in the current study, that study also found an association between women, without social support and with polypharmacy, among other health conditions, with this low performance.

Differences in physical performance have been shown in relation to the level of physical activity. Latin America has perhaps one of the highest proportions of inactive older persons [36,37] and this, in turn, is related to performance, autonomy, and functional capacity. A study conducted in 2020 [38] found that active older persons had minimal performance limitations at 65% while sedentary older persons had moderate limitations, amounting to 81.8%.

An interesting aspect is related to the presence of the female population, aged between 60 and 70 years who live alone in self-owned homes: this is considered their closest environment. In Brazil in particular, since 1940, housing policies have been intensified to promote access to home ownership. These policies seem to have contributed to the percentage of rented housing in Brazil being considerably lower than in more developed countries such as France, where this percentage corresponded to 37% of residential properties [39,40], while in Brazil during 2010, it was at 18.32% [41], and at 17.9% in 2015 [42]. Specifically, in Aveiro, Brazil, 56.8% of older persons live in self-owned homes, and among them, a higher frequency of moderate physical performance was found [43]. It has already been shown that older people with their own homes can make greater adaptations to favor their autonomy [44].

On the other hand, in the Netherlands, in 2018, 94.6% of those over 65 years of age lived independently, and in 2015, 73.4% lived in multi-story buildings, 16.9% in a house adapted for older persons, and 41.9% of people lived in social housing [45]. An aspect to highlight in all these studies is the need for more social housing, which guarantees older tenants’ long-term tenure, or mechanisms that help them obtain self-owned homes. This favors physical adaptations, which results in their well-being and autonomy during their lifetime [46].

## 5. The Social and Physical Environment

The age-friendly communities approach is rooted in the ecological theory of aging. It emphasizes the interconnection between social and physical environments to determine the health, well-being, and capacity of adults to successfully age and contribute to their communities [47,48]. It is known that both the physical and social environments affect the well-being of older persons. Good governance and comprehensive planning are fundamental [49] for lifelong well-being, as they help people to remain independent for as long as possible and provide care and protection when necessary, respecting the autonomy and dignity of older persons. This leads to the improvement of their health and social inclusion [50,51].

A study on older persons in France revealed that those who lived in more favorable geographical environments tended to go out more often than those who lived in environments with higher geographical barriers [52]. In Mooca, Brazil, older people highlighted that the factors that hinder or favor this condition of access to educational establishments and activities are related to proximity, admission criteria, information, communication, affordability, safety, and interest [53].

Our findings show that our population reported higher physical performance when a better social environment related to the space of their home and attributes of the physical environment was reported. All of these are elements that, together, facilitate greater autonomy both inside and outside of their homes. The environment is the place where individuals develop certain life conditions. It includes the interaction between the physical, social, and cultural characteristics in which the individual lives and, therefore, interacts with others and with institutions on a regular basis, denoting their participation and autonomy [54,55]. However, progress in planning strategies to make age-friendly cities in Latin American countries such as Colombia remains a challenge [56].

Despite the WHO’s proposed strategy of Age-Friendly Global Cities to improve the environmental, social, and economic factors that influence the well-being of older adults [2], to date, in Colombia, only one provincial capital (Ibagué) [57] has achieved its implementation. Medellín has tried to advance through a Public Policy on Aging and Old Age that extends to 2027 [58]. Based on the diagnosis made of the city’s aged population, unsatisfied basic needs were detected (for example, lack of drinking water). This changes the priorities of care and intervention to focus specifically on solving problems related to health insurance, education, and the generation of productive projects. However, within this diagnosis, it has been reported that older persons in the city perceived that several of the existing spaces were in poor condition, and others still needed to improve accessibility for the population with reduced mobility. In others, inappropriate use or a lack of use by the community was observed. The strategies boil down to improving access to public transport, without evidence of a strategy that includes real investment in improvements for the physical and social infrastructure.

Nationwide, according to the Health and Wellness Survey of Older Persons, SABE, at least one in four older persons reports that, in their neighborhood of residence, there are many irregular sidewalks and that they do not have access to public transport near their home. Four out of ten said that there are no parks, walking areas, sports, or recreation centers. More than half consider that there are no places to sit or rest at bus stops or in parks and that there are no public transport options for people with disabilities, or that they do not have adequate parking lots, which limits their participation and operation [12].

Aging in the “right” place means the ability to live in the place that best suits a person’s needs and preferences, which may or may not be one’s own home [8]. Demographic aging creates a demand for residential and care services different from those currently available. It is required that the real estate and care sector adapt accordingly to this new reality [59] and to the global trend of aging at home and not in retirement homes. Moving to another home generates feelings of loss of empowerment, independence, and autonomy, and one must adhere to new routines, new companies, and new caregivers [60]. It is critical for healthy aging with autonomy to have diverse and innovative housing options available for people to move within their community [61]. This should offer housing options with universal design features, supported by services that provide better environments for aging [48,62,63,64,65].

Other studies have pointed to the complex and multifactorial relationship between housing and health in older persons [65]. There have been multiple studies testing interventions to enable an age-in-place process. These interventions focus on adaptations within the home, such as handrails, bathroom modifications, and non-slip steps, which can further help seniors age in their homes [66]. In-home modifications can also reduce the dangers that can lead to the hospitalization of seniors [67]. Home care and technological devices (such as panic buttons) can also reduce injuries and other risks to allow older persons to age in place, maintain their independence for longer, and improve self-confidence and coping strategies [68,69].

An indirect consequence of the low prioritization in the improvement of social and physical environments has to do with age discrimination. Although the progress made in relation to the social situation and legal status of older citizens in the population cannot be ignored and age is a protected characteristic in the legislation of most countries, these do not have as high a profile as aspects related to gender, religion, and ethnicity. Work must continue so that in popular narratives, aging is not presented in a stereotypical or negative way [51,70]. As age advances, it seems that older persons are left behind in their homes or places of housing, with less functionality and social participation. Understanding the impact of the environments in which these age groups live and participate is particularly relevant since changes inherent in old age make them more vulnerable to the detrimental effects of overcoming other conditions.

There is evidence of the need to understand how environmental changes affect aging people, how the life stories of aging individuals connect with the social and spatial history of the urban environment, and how the stability of the neighborhood and residence time of an individual’s residence in a specific place influences health. Building this evidence will provide both researchers and policymakers with tools to respond to the public health challenges of aging and urbanization in the coming decades [71].

## 6. The Built Environment

Unlike the results of the present study, most people aged 65 and over in the Netherlands are generally satisfied with the different facilities in their neighborhood, such as shops for daily use and public transport (85%), public lighting (83%), green areas (76%) and maintenance of roads, and bike paths (71%). People are less satisfied with parking (41%). In addition, 19.7% of people indicate that they sometimes feel unsafe in their neighborhood. Within the physical space and residential areas, there are health centers with physiotherapists and general practitioners. In addition, these centers have house services, and additional services focused on financial matters related to the filling of tax forms and debt advice. Having all these available under one roof, in an accessible building with automated doors and level access, makes it easier for seniors in the Netherlands to make use of them [45].

In Colombia, as well as in several Latin American countries, environments built for older persons present challenges [72], mainly because changes inherent in old age make them more vulnerable to the detrimental effects of socially and physically deficient environments. Steep streets, uneven surfaces, absence of pedestrian-friendly infrastructure, low compliance with traffic rules, lack of pedestrian devices, and dirty and unsafe streets among other barriers and hazards are common [56,73]. Therefore, some of these barriers, along with their families, restrict the participation of older adults in neighborhoods and in the city. In more developed countries, there are probably resources and investment in infrastructure to facilitate free and external recreational spaces not only for older persons but for families, generating a better perception of care and opportunities for the older person. In Latin countries, this has not yet become a priority.

In a study on the effect of the proposed Age-Friendly Cities on their satisfaction with life, it was highlighted that there is a lack of knowledge of public aid for the rehabilitation and adaptation of housing. There is a need to improve information on these aspects, along with a simplification of procedures, since many old homes need to be adapted (installation of elevators, ramps, and replacement of bathtubs with showers). Among the aspects they mention, they highlight the need to improve the information provided on policies that consider the longevity of society and services aimed at promoting the independence and autonomy of seniors, allowing them to continue living in their own homes for longer [74]. Priority should be given to the creation of more compact urban forms and the consolidation of housing adapted to older persons, capitalizing on existing services and facilities [75].

## 7. Limitations

One of the limitations of this study lies in its design and the representability of the data. The cross-sectional epidemiological designs present several limitations related to the fact that the causality or temporality of the associated factors (environmental surroundings) and their effect on the outcome (functional autonomy) cannot be estimated. There is a limitation due to reverse causality since the precedency of exposures cannot be determined. In this sense, the functional autonomy of individuals may be what determines the perceptions they have about the environment, so this association should be treated with caution. Likewise, some of the empirical evidence that has been related comes from cross-sectional designs that keep the same bias. Regarding representativeness, a probabilistic, random cluster and two-stage sampling were performed.

## 8. Conclusions

With increasing population aging and urbanization, the development of age-friendly environments for older persons needs to be prioritized by researchers and policymakers. Thinking about better social and physical facilities is beneficial for all people, regardless of age. Exploring how the physical and social environments surrounding housing are associated with the functional performance of older persons can generate useful information to support public health and city infrastructure strategies that improve their physical performance and maintain their autonomy of action.

Within existing public policy, one of its principles is to prioritize autonomy. The understanding of environments as part of complex outcomes is a priority not only in the functional capacity but also in the health and well-being of older persons. Therefore, their study, as well as the creation of strategies that prioritize environmental conditions, should be part of regional and national agendas and investments. This will make a public health contribution to future generations, allowing them to count on age-friendly environments.

## Figures and Tables

**Table 1 ijerph-20-00409-t001:** Proportional distribution of older persons according to individual conditions, classified by functional autonomy of action (SPPB). Urban area of the city of Medellín, Colombia, 2021.

Individual Conditions	Functional Autonomy										
	Moderate (157)		Very Low/Low (18)		Total (175)		X^2^	*p*-Value	cOR	95% CI	
	*n*	%	*n*	%	*n*	%				Ll	Ul
Sex											
Man	56	35.7	7	38.9	63	36.0	0.07	0.79	0.87	0.32	2.37
Woman	101	64.3	11	61.1	112	64.0			1.00		
Age											
≤70 years	111	70.7	9	50.0	120	68.6	3.07	0.08	2.41	0.90	6.47
70+ years	46	29.3	9	50.0	55	31.4			1.00		
Marital status											
No partner	85	54.1	14	77.8	99	56.6	3.40	0.07	0.34	0.11	1.07
With a partner	72	45.9	4	22.2	76	43.4			1.00		
Type of housing											
Owned	92	58.6	6	33.3	98	56.0	9.00	0.01	5.37	1.63	17.69
Rented	45	28.7	5	27.8	50	28.6			3.15	0.89	11.14
Family	20	12.7	7	38.9	27	15.4			1.00		
Familiar functionality											
Functional	101	64.3	12	66.7	113	64.6	0.04	0.84	0.90	0.32	2.53
Dysfunctional	56	35.7	6	33.3	62	35.4			1.00		
Poor social support											
Yes	1	0.6	1	5.6	2	1.1	2.38	0.12	0.11	0.01	1.82
No	156	99.4	17	94.4	173	98.9			1.00		
Rejection in health institutions											
Yes	14	8.9	4	22.2	18	10.3	2.87	0.09	0.34	0.10	1.18
No	143	91.1	14	77.8	157	89.7			1.00		

cOR: crude Odds Ratio; X^2^: Chi-square hypothesis test; 95% CI: 95% confidence interval; Ll, lower limit; Ul: upper limit.

**Table 2 ijerph-20-00409-t002:** Proportional distribution of older persons according to the perception of the social environment, classified by functional autonomy of action (SPPB). Urban area of the city of Medellín, Colombia, 2021.

Social Environment	Functional Autonomy										
	Moderate (157)		Very Low/Low (18)		Total (175)		X^2^	*p*-Value	aOR	95% CI	
	*n*	%	*n*	%	*n*	%				Ll	Ul
Living space							0.49	0.49			
Yes	131	83.4	13	72.2	144	82.3			1.70	0.38	7.50
No	26	16.6	5	27.8	31	17.7			1.00		
Quiet neighborhood							0.00	0.98			
Yes	152	96.8	17	94.4	169	96.6			1.03	0.07	14.73
No	5	3.2	1	5.6	6	3.4			1.00		
Social activities in the neighborhood							0.45	0.50			
Yes	61	38.9	9	50.0	70	40.0			0.22	0.00	18.93
No	96	61.1	9	50.0	105	60.0			1.00		
Event schedule							0.05	0.82			
Yes	62	39.5	9	50.0	71	40.6			0.19	0.00	Ind
No	95	60.5	9	50.0	104	59.4			1.00		
Event facilities							0.01	0.90			
Yes	65	41.4	9	50.0	74	42.3			2.43	0.00	Ind
No	92	58.6	9	50.0	101	57.7			1.00		
Community partnerships							0.00	0.96			
Yes	74	47.1	9	50.0	83	47.4			1.10	0.03	34.58
No	83	52.9	9	50.0	92	52.6			1.00		
Intergenerational activities							0.74	0.39			
Yes	73	46.5	8	44.4	81	46.3			5.27	0.12	230.20
No	84	53.5	10	55.6	94	53.7			1.00		
Sharing knowledge and experiences							0.16	0.69			
Yes	70	44.6	8	44.4	78	44.6			0.52	0.02	13.09
No	87	55.4	10	55.6	97	55.4			1.00		
Respect and recognition							0.06	0.80			
Yes	63	40.1	7	38.9	70	40.0			1.36	0.12	15.14
No	94	59.9	11	61.1	105	60.0			1.00		
Urban facilities							1.96	0.16			
Yes	65	41.4	6	33.3	71	40.6			0.19	0.02	1.94
No	92	58.6	12	66.7	104	59.4			1.00		
Absence in recreational areas							1.20	0.27			
Yes	61	38.9	5	27.8	66	37.7			3.78	0.35	40.87
No	96	61.1	13	72.2	109	62.3			1.00		
Community action boards (JAC) information							3.35	0.07			
Yes	61	38.9	4	22.2	65	37.1			11.28	0.84	150.99
No	96	61.1	14	77.8	110	62.9			1.00		
Job offers							0.93	0.33			
Yes	59	37.6	5	27.8	64	36.6			0.30	0.03	3.41
No	98	62.4	13	72.2	111	63.4			1.00		
Health campaigns							0.49	0.49			
Yes	131	83.4	13	72.2	144	82.3			1.70	0.38	7.50
No	26	16.6	5	27.8	31	17.7			1.00		

aOR: adjusted Odds Ratio; X^2^: Chi-square hypothesis test; 95% CI: 95% confidence interval; Ll, lower limit; Ul: upper limit; Und: indeterminate. Note: Logistic regression model adjusted for sex, age, marital status, and type of housing (R^2^ = 0.26).

**Table 3 ijerph-20-00409-t003:** Proportional distribution of older persons according to perception of the physical environment, classified by functional autonomy of action (SPPB). Urban area of the city of Medellín, Colombia, 2021.

Physical Environment	Functional Autonomy										
	Moderate (157)		Very Low/Low (18)		Total (175)		X^2^	*p*-Value	aOR	95% CI	
	*n*	%	*n*	%	*n*	%				Ll	Ul
Visual pollution							1.93	0.16			
Yes	51	32.5	7	38.9	58	33.1			0.29	0.05	1.66
No	106	67.5	11	61.1	117	66.9			1.00		
Noise pollution							0.06	0.81			
Yes	74	47.1	5	27.8	79	45.1			1.95	0.01	426.85
No	83	52.9	13	72.2	96	54.9			1.00		
Adequate public toilets							0.19	0.67			
Yes	70	44.6	4	22.2	74	42.3			4.07	0.01	2370.08
No	87	55.4	14	77.8	101	57.7			1.00		
Equipment of common areas							0.02	0.88			
Yes	64	40.8	4	22.2	68	389			0.79	0.04	17.85
No	93	59.2	14	77.8	107	61.1			1.00		
Park lighting							3.68	0.06			
Yes	68	43.3	6	33.3	74	42.3			19.97	0.94	425.42
No	89	56.7	12	66.7	101	57.7			1.00		
Wide sidewalks							4.84	0.03			
Yes	63	40.1	8	44.4	71	40.6			0.06	0.01	0.74
No	94	59.9	10	55.6	104	59.4			1.00		
Unobstructed sidewalks							0.21	0.64			
Yes	56	35.7	7	38.9	63	36.0			0.48	0.02	10.54
No	101	64.3	11	61.1	112	64.0			1.00		
Traffic light timer							0.02	0.90			
Yes	48	30.6	6	33.3	54	30.9			0.80	0.03	24.21
No	109	69.4	12	66.7	121	69.1			1.00		
Informative signage							0.60	0.44			
Yes	50	31.8	6	33.3	56	32.0			0.21	0.00	11.40
No	107	68.2	12	66.7	119	68.0			1.00		
Access to housing and buildings							0.00	0.99			
Yes	49	31.2	5	27.8	54	30.9			1.02	0.03	39.51
No	108	68.8	13	72.2	121	69.1			1.00		
Preferred rows							0.32	0.57			
Yes	47	29.9	3	16.7	50	28.6			3.52	0.04	278.20
No	110	70.1	15	83.3	125	71.4			1.00		
Parks with rest chairs							2.57	0.11			
Yes	48	30.6	5	27.8	53	30.3			0.02	0.00	2.31
No	109	69.4	13	72.2	122	69.7			1.00		
Preferential fares on public transport							3.95	0.05			
Yes	51	32.5	3	16.7	54	30.9			90.33	1.06	7676.85
No	106	67.5	15	83.3	121	69.1	0.02	0.88	1.00		

aOR: adjusted Odds Ratio; X^2^: Chi-square hypothesis test; 95% CI: 95% confidence interval; Ll, lower limit; Ul: upper limit; Und: indeterminate. Note: Logistic regression model adjusted for sex, age, marital status, and type of housing (R^2^ = 0.331).

**Table 4 ijerph-20-00409-t004:** Proportional distribution of older people according to the perception of the built environment, classified by functional autonomy of action (SPPB). Urban area of the city of Medellín, Colombia, 2021.

Social Environment	Functional Autonomy										
	Moderate (157)		Very Low/Low (18)		Total (175)		X^2^	*p*-Value	OR	95% CI	
	*n*	%	*n*	%	*n*	%				Ll	Ul
Street separators											
Yes	23	14.6	4	22.2	27	15.4	0.69	0.40	0.60	0.18	1.99
No	134	85.4	14	77.8	148	84.6			1.00		
Arborization											
Yes	55	35.0	6	33.3	61	34.9	0.02	0.89	1.08	0.39	3.03
No	102	65.0	12	66.7	114	65.1			1.00		
Street cleanness											
Yes	24	15.3	3	16.7	27	15.4	0.02	0.88	0.90	0.24	3.36
No	133	84.7	15	83.3	148	84.6			1.00		
Graffiti											
Yes	42	26.8	7	38.9	49	28.0	1.16	0.28	0.58	0.21	1.58
No	115	73.2	11	61.1	126	72.0			1.00		
Murals											
Yes	52	33.1	6	33.3	58	33.1	0.00	0.99	0.99	0.35	2.79
No	105	66.9	12	66.7	117	66.9			1.00		
Public transport											
Yes	154	98.1	16	88.9	170	97.1	3.83	0.05	6.42	0.99	41.29
No	3	1.9	2	11.1	5	2.9			1.00		
Good track conditions											
Yes	12	7.6	5	27.8	17	9.7	6.43	0.010	0.22	0.07	0.71
No	145	92.4	13	72.2	158	90.3			1.00		
Adequate lighting											
Yes	11	7.0	2	11.1	13	7.4	0.39	0.53	0.60	0.12	2.96
No	146	93.0	16	88.9	162	92.6			1.00		
High vehicular flow											
Yes	138	87.9	14	77.8	152	86.9	1.39	0.24	2.08	0.62	6.96
No	19	12.1	4	22.2	23	13.1			1.00		
High speed of vehicles											
Yes	150	95.5	16	88.9	166	94.9	1.36	0.24	2.68	0.51	14.00
No	7	4.5	2	11.1	9	5.1			1.00		
Speed control											
Yes	13	8.3	5	27.8	18	10.3	5.82	0.02	0.24	0.07	0.76
No	144	91.7	13	72.2	157	89.7			1.00		
Devices for street crossing											
Yes	10	6.4	4	22.2	14	8.0	4.81	0.03	0.24	0.07	0.86
No	147	93.6	14	77.8	161	92.0			1.00		
Passersby											
Yes	26	16.6	4	22.2	30	17.1	0.36	0.55	0.70	0.21	2.28
No	131	83.4	14	77.8	145	82.9			1.00		
Cycle routes											
Yes	21	13.4	4	22.2	25	14.3	1.01	0.32	0.54	0.16	1.80
No	136	86.6	14	77.8	150	85.7					
Good condition of the sidewalks											
Yes	55	35.0	7	38.9	62	35.4	4.72	0.32	0.45	0.13	1.61
No	8	5.1	2	11.1	10	5.7			0.23	0.04	1.45
Under repair	21	13.4	5	27.8	26	14.9			0.24	0.06	0.98
Partly	3	1.9	0	0.0	3	1.7			Und	Und	Und
No sidewalks	70	44.6	4	22.2	74	42.3			1.00		
Continuous sidewalks											
Yes	63	40.1	7	38.9	70	40.0	8.66	0.01	0.48	0.13	1.72
No	19	12.1	7	38.9	26	14.9			0.15	0.04	0.55
No sidewalks	75	47.8	4	22.2	79	45.1			1.00		
Road steepness											
Very steep	37	23.6	4	22.2	41	23.4	2.78	0.25	1.62	0.47	5.56
Moderately	63	40.1	4	22.2	67	38.3			2.76	0.82	9.30
Lightly	57	36.3	10	55.6	67	38.3			1.00		
Number of lanes											
1	104	66.2	9	50.0	113	64.6	1.35	0.85	Und	Und	Und
2	22	14.0	4	22.2	26	14.9			Und	Und	Und
3	20	12.7	2	11.1	22	12.6			Und	Und	Und
4	11	7.0	1	5.6	12	6.9			Und	Und	Und
5	0	0.0	2	11.1	2	1.1			1.00		

OR: Odds Ratio; X^2^: Ch squared hypothesis test; 95% CI: 95% confidence interval; Ll, lower limit; Ul: upper limit; Ind: indeterminate. Note: Simple logistic regression model.

## Data Availability

Cardona, D.; Segura, A.M.; Segura, D.A.; Robledo, C.A.; Muñoz, D.I. *Salud y Bienestar Mental de la Pewrsona Mayor en cinco Ciudades de Colombia (2020–2022)*; Editorial CES: Medellín, Colombia, 2022; ISBN: 978-958-5101-62-3.

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
