# Peer review of "Friendly Residential Environments That Generate Autonomy in Older Persons"

_ijerph, 2022, doi:10.3390/ijerph20010409_

Round 1

Reviewer 1 Report (Previous Reviewer 2)

I have read the new version of the manuscript and I think the authors have adequately addressed my questions.

This manuscript is a resubmission of an earlier submission. The following is a list of the peer review reports and author responses from that submission.

Round 1

Reviewer 1 Report

Summary: it is unclear what type of study the authors conducted, survey, interview, or others.

Introduction: The information is relevant. However, the structure and content can be more cohesive. Also, the sentences are too long, preventing reviewers from understanding. For example, the paragraph from lines 80 to 88 is just one sentence. Plus, there are typos and grammar errors in the text, e.g., the column in line 63 should be removed. At the end of the introduction, the purpose of the study needs to be specified to let reviewers know what this study tries to solve. 

Methods: the methods need further clarification. For example, it is helpful to introduce SPPB in more detail, e.g., what does this tool measure and how the results of SPPB could be interpreted? In addition, for the statistical analysis, it is necessary to explain what correlation/relationship you would like to study using each type of statistical analysis. Moreover, the author mentioned hypothesis tests in the method; however, what the hypotheses are is not clear. 

Results: The result should be further elaborated. For example, the authors need to help reviewers understand what PR means, its implication, and what it tells us with p-value, Xand CI. It is hard to understand as it is.

Author Response

The authors of the referenced manuscript, appreciate your comments and recommendations for the improvement of a version more adjusted to those expected by the International Journal of Environmental Research and Public Health and therefore we respond to each of them:
Comments and Suggestions
Answers
Summary: it is unclear what type of study the authors conducted, survey, interview, or others.
The observation is met, and the type of study is adjusted in the abstract and in materials, according to the authors Celentano D & Szklo M. Gordis Epidemiology, 6ª ed. Philadelphia, PA: Elsevier; 2019
Introduction: The information is relevant. However, the structure and content can be more cohesive. Also, the sentences are too long, preventing reviewers from understanding. For example, the paragraph from lines 80 to 88 is just one sentence. Plus, there are typos and grammar errors in the text, e.g., the column in line 63 should be removed.

The wording is revised, and the suggested paragraphs are adjusted.
At the end of the introduction, the purpose of the study needs to be specified to let reviewers know what this study tries to solve. 
The purpose of the study is included in the abstract and at the end of the introduction
Methods: the methods need further clarification. For example, it is helpful to introduce SPPB in more detail, e.g., what does this tool measure and how the results of SPPB could be interpreted?

Within methods, we added the item: "Dependent variable: Functional autonomy (SPPB)" to explain what the tool measures and support the interpretation of the findings.
In addition, for the statistical analysis, it is necessary to explain what correlation/relationship you would like to study using each type of statistical analysis. Moreover, the author mentioned hypothesis tests in the method; however, what the hypotheses are is not clear. 
The methods component is adjusted, in the abstract and in the manuscript, where it clarifies: "Univariate and bivariate analyses were performed with these variables, where proportion ratio (PR), association hypothesis test, and confidence intervals were estimated, using logistic regression models"
Results: The result should be further elaborated. For example, the authors need to help reviewers understand what PR means, its implication, and what it tells us with p-value, Xand CI. It is hard to understand as it is.

The reviewer's observation is accepted, the acronyms are clarified, and their interpretation is improved

Reviewer 2 Report

Thank you for allowing me to read your study. It is interesting as it addresses an important aspect of healthy ageing such as the social and built environment. However, it presents some limitations that must be corrected.

General comment:

Throughout the text and title, please, change the term “elderly” to “older people” or “older persons”, as recommended by the World Health Organization.

Also, the writing should be reviewed, as there are sentences too long or incomplete.

Abstract:

- The text in Methods “which were classified with moderate and very low/low performance” is also in Results. If it is a result of the study, and not an inclusion criterion, the information should be removed from the Methods.

- Please, add quantitative results to the Abstract, particularly those related to the logistic regression model.

Materials and methods:

- Please, add details (a description) on the included variables.

- Which were the cut-off scores used for the SPPB scale to classify the sample as having or not functional limitations?

- Please, include details on the logistic regression models. What were the dependent and independent/control variables?

Results:

- The initials PR (prevalence ratio) are incorrectly spelled several times throughout the Results section.

- The text in p. 4 “Of these conditions, those who contribute most to autonomy were age between 60 and 70 years (RP=2,413), women, married or in a free union, homeowners (PR=5,370) and do not perceive rejection in health services.” is not accurate as authors are performing a bivariate analysis in Table 1. Moreover, the differences are not significant. No conclusions on causality or even in terms of determinants can be drawn here. The same happens in the sentence in p. 5 “in city adaptations it was also found that they could favor autonomy (moderate or high) if there are spaces to share with people of other ages”. Please, revise the text in Results accordingly.

- Please, correct the sentence “When this perception is positive, there is evidence of an increase in the probability of having a moderate functional autonomy of action,….” This is not a longitudinal study, so no increases or decreases of rates or percentages could be seen.

- In Methods, authors state they have performed logistic regression models but they are not displayed in Results.

- Please, do not discuss the results in this section, but in Discussion.

Discussion:

- The Discussion is hampered by the absence of a regression model that controls by possible confounding variables and interactions (for example, age and sex). Moreover, most associations with functioning in the bivariate analyses are not significant, probably due to the relatively small sample in the low/very low functional autonomy group.

- The Discussion lacks an explanation of the study results in the context of Medellin or Colombia.

- Please, include a section on limitations of the study.

Tables:

- In table 2, please, explain the meaning of JAC Information.

Author Response

The authors of the referenced manuscript, appreciate your comments and recommendations for the improvement of a version more adjusted to those expected by the International Journal of Environmental Research and Public Health and therefore we respond to each of them:
Comments and Suggestions
Answers
Thank you for allowing me to read your study. It is interesting as it addresses an important aspect of healthy ageing such as the social and built environment. However, it presents some limitations that must be corrected.
Thank you for your review and feedback, which allowed us to improve the manuscript
Throughout the text and title, please, change the term “elderly” to “older people” or “older persons”, as recommended by the World Health Organization.
The suggestion was followed, and the word "elderly" was removed from the entire document.
Also, the writing should be reviewed, as there are sentences too long or incomplete.
The suggestion is followed
Abstract:
- The text in Methods “which were classified with moderate and very low/low performance” is also in Results. If it is a result of the study, and not an inclusion criterion, the information should be removed from the Methods.
- Please, add quantitative results to the Abstract, particularly those related to the logistic regression model.
The reviewer's suggestion is followed, and the operationalization of the dependent variable is adjusted. Other results are also included in the abstract
Materials and methods:
- Please, add details (a description) on the included variables.
- Which were the cut-off scores used for the SPPB scale to classify the sample as having or not functional limitations?
- Please, include details on the logistic regression models. What were the dependent and independent/control variables?
The SPPB cut-off points used were
0 to 6: Low or very low performance
7 to 9: Moderate performance
From 10 to 12: High performance (satisfactory)
These were clarified in the method item entitled: "Dependent variable: Functional autonomy (BPPS)" and are duly referenced

The observation is followed, and the acronym is unified in the document. In cross-sectional studies, the prevalence ratio or proportions ratio (PR), a measure used in this manuscript, is usually used, according to Schiaffino A et al. Odds ratio is used in other cross-sectional studies. Gac Sanit 2003;17(1):70-4.

Results:
- The initials PR (prevalence ratio) are incorrectly spelled several times throughout the Results section.
- The text in p. 4 “Of these conditions, those who contribute most to autonomy were age between 60 and 70 years (RP=2,413), women, married or in a free union, homeowners (PR=5,370) and do not perceive rejection in health services.” is not accurate as authors are performing a bivariate analysis in Table 1. Moreover, the differences are not significant. No conclusions on causality or even in terms of determinants can be drawn here. The same happens in the sentence in p. 5 “in city adaptations it was also found that they could favor autonomy (moderate or high) if there are spaces to share with people of other ages”. Please, revise the text in Results accordingly.
The observations are followed, and the hypothesis tests (X2) and the strength of the association (PR) are adjusted.
- Please, correct the sentence “When this perception is positive, there is evidence of an increase in the probability of having a moderate functional autonomy of action,….” This is not a longitudinal study, so no increases or decreases of rates or percentages could be seen.
The wording of the sentence is adjusted
- In Methods, authors state they have performed logistic regression models but they are not displayed in Results.
- Please, do not discuss the results in this section, but in Discussion.
It was clarified that logistic regression was used for the calculation of raw and adjusted PR.
Discussion:
- The Discussion is hampered by the absence of a regression model that controls by possible confounding variables and interactions (for example, age and sex). Moreover, most associations with functioning in the bivariate analyses are not significant, probably due to the relatively small sample in the low/very low functional autonomy group.
- The Discussion lacks an explanation of the study results in the context of Medellin or Colombia.
The suggestion is accepted and in the development of the discussion, the context of the urban policy of Medellín for aging in terms of the environment in the local and national context is clarified.
- Please, include a section on limitations of the study.
A limitations section was included
Tables:
- In table 2, please, explain the meaning of JAC Information.
The term Community Action Board (JAC) is adjusted

Reviewer 3 Report

Overall Comments:

This study reports on the associations between a number of perceived environmental domains and functional autonomy among older residents of Medellin, Colombia. The topic, identifying environmental features that might enhance the ability of aging populations to maintain autonomy and age in place, is an important one. I am not sure that this small cross sectional study can be used to address the question given the potential for reverse causality (that the functional autonomy of individuals determines perceptions of the environment). Further, the sampling frame (of neighborhoods, blocks, and finally individuals), may contribute to bias in the determination of the prevalence of functional status and in the relationship between environmental features and functional status. Finally, the authors seem to be confused about what measures of association have been presented in each table, and in the text, and indeed there is no evidence that any adjusted regression analyses were performed. I think with some substantial revisions to the introduction, methods, results and discussion this article could make an important contribution to the literature about environments and healthy aging in Latin America.

Specific remarks:

1.      Abstract (line 26): a prevalence ratio is not a standard outcome for a logistic regression analysis- in fact, all results presented in the manuscript appear to be odds ratios. Please confirm that prevalence ratios were calculated or correct this.

2.      Abstract (lines 33-34): this statement (health outcomes…that underlie it) does not align with this study. No interaction was evaluated, no multi-level analysis was performed, and indeed no specific consideration of geographic scale was evaluated. This should be removed here and in the discussion (lines 431-432).

3.      The introduction is too long and insufficiently focused. The introduction should be a brief summary of the public health concern and specific considerations that builds to the study question. I would recommend radically shortening the introduction (perhaps by cutting lines 46-53, 67-70, and 90-100).

4.      Sampling methodology is not sufficiently explained (lines 132-134): we need to know how many neighborhoods there were in the sampling frame (and how many ended up in the sample), how many blocks (median, minimum and maximum) in all neighborhoods (and then how many were selected in neighborhoods that ended up in the sample), and how many individuals were selected per block. Given the sampling methodology (line 133), the analysis likely should have used statistical weights to ensure that the survey responses were representative of the target population (older community dwelling adults in Medellin).

5.      It is unclear what absolute and relative frequency measures are (line 142-143): relative to what? Response frequencies were calculated, overall and in strata of functional status.

6.      What statistical measures were used in the estimation of physical performance (line 144-145)? Does this mean a regression analysis was done to determine physical performance? Or did the authors generate scores for the SPPB based on a previously-published method?

7.      Hypothesis tests aren’t calculated, they are performed (lines 146-147).

8.      All bivariate associations reported in the tables are odds ratios, not prevalence ratios (line 150).

9.      No results are reported for the binary logistic regression (lines 150-151): This should either be removed from the manuscript or the results should be presented. In fact, it may benefit the authors to eliminate the unadjusted odds ratios presented in Tables 2-4 and instead present odds ratios adjusted for sex, age, marital status, housing type.

10.   Language throughout the manuscript occasionally lapses into causal language (example line 152: “contribute to” should be “are associated with”; line 290: “influence this outcome” should be “are associated with”). This needs to be corrected throughout the manuscript.

11.   Methods: a detailed description of the study variables and response options needs to be provided. The reader will be confused when they get to Table 1 and see familiar functionality/having been rejected by a health institution without any explanation of what is being reported.

12.   Table 1 (and 2-4): the presentation of frequencies and unadjusted odds ratios (no prevalence ratios are presented) in not necessary. The frequencies alone are sufficient (and the p-value for the chi-square test needs to indicate that this is for the test comparing moderate to low/very low functional autonomy- can be done in a footnote). I would recommend the authors replace the unadjusted ORs in Tables 2-4 and replace them with adjusted ORs from logistic regression models (see comment 9).

13.   Tables 1-4: please keep only 2 decimal places for all measures of association (odds ratios and confidence intervals).

14.   Results: the authors have repeatedly included parenthetical reference to “(RP=)”. I assume this is supposed to be PR=. Please correct throughout.

15.   Results: the authors intermittently report survey respondent perception of the environment as if they were directly and objectively assessed by the researchers (e.g. line 220: in public parks there are not clean…). All statements of results based upon participant reported variables need to begin with “Participants reported…”.

16.   Discussion: Line 288- the statement that the elderly of Medellin have moderate physical performance can’t really be supported by this data if the analysis was not weighted to the sampling frame.

17.   Line 308: “what does none of them reached a satisfactory level” mean? Does this mean that their physical performance does not meet some fully functional status? This should be explained in the methods with regards to the scoring of the SPPB.  

18.   Lines 309-330: This section of the discussion fails to acknowledge the potential for reverse causality. The direction of the association (self-owned homes leading to higher performance or higher performance leading to greater likelihood of living in self-owned home in old age).

19.   Line 354: “performance has a better…”, needs to be replaced with “performance reports a better…”.

20.   Lines 374-375: “However,…existing communities” is not a full sentence. What is this supposed to be communicating?

21.   Lines 390-396: Age discrimination is very abruptly introduced here. How does this apply: is it in planning for the needs of the elderly are subordinated to other groups?

22.   Lines 405-415: This could use some more analysis- why is the situation in the Netherlands different from Colombia? Is it surprising that a high-income, densely populated, and not newly older (having had decades of experience with an aging population) country like the Netherlands may do a better job of meeting the needs of its older individuals?

23.   Line 431: “elderly is **associated with** physical social,…”.

24.   Lines 432-434: This sentence doesn’t represent the value of this research (in fact, it overstates it). Exploring how environments are associated with performance may generate information to support design strategies to improve performance and maintain autonomy. The exploration of the environments does not in itself contribute to design strategies to improve performance and generate autonomy.

Author Response

The authors of the referenced manuscript, appreciate your comments and recommendations for the improvement of a version more adjusted to those expected by the International Journal of Environmental Research and Public Health and therefore we respond to each of them:
Orden
Comments and Suggestions
Answers
This study reports on the associations between a number of perceived environmental domains and functional autonomy among older residents of Medellin, Colombia. The topic, identifying environmental features that might enhance the ability of aging populations to maintain autonomy and age in place, is an important one.
The words of the reviewer are appreciated, we also consider that these cross-sectional studies are important, as they allow to build hypotheses for future studies
I am not sure that this small cross sectional study can be used to address the question given the potential for reverse causality (that the functional autonomy of individuals determines perceptions of the environment).
The authors share the concern of reverse causality and therefore it is recognized as a limitation of the study and it is clarified in the methods that it was not sought to determine causality, since the design does not allow it.
Further, the sampling frame (of neighborhoods, blocks, and finally individuals), may contribute to bias in the determination of the prevalence of functional status and in the relationship between environmental features and functional status.
The information on the sampling was expanded, it was probabilistic random two-stage clusters: the selection of the neighborhoods was made within each of the 16 communes of Medellín, with systematic random sampling three neighborhoods per commune were selected for a total of 51 neighborhoods of the city,  as a secondary sampling unit (SMU) and within each neighborhood, three blocks were selected as the primary sampling unit (PSU) by simple random sampling, for a total of 175 blocks and there, an elderly person residing in the block was randomly taken, as a final sampling unit (FSU), in whom the residential environments (physical and social) was investigated.
Finally, the authors seem to be confused about what measures of association have been presented in each table, and in the text, and indeed there is no evidence that any adjusted regression analyses were performed.
In cross-sectional studies, the prevalence ratio or proportion ratio, a measure used in this manuscript, is usually used, according to Schiaffino A et al. Odds ratio is use in cross-sectional studies. Gac Sanit 2003;17(1):70-4.
The chi-square hypothesis test (X2) was also used as a measure of association with its p-value, as shown in the tables. In the new version of the manuscript it is clarified that logistic regression was used for the calculation of crude and adjusted PR measures.
I think with some substantial revisions to the introduction, methods, results and discussion this article could make an important contribution to the literature about environments and healthy aging in Latin America.
We thank the reviewer for his sincere comments and hope to make a theoretical contribution; Therefore, adjustments have been made to the summary, introduction, methods, results and discussion.
1
Abstract (line 26): a prevalence ratio is not a standard outcome for a logistic regression analysis- in fact, all results presented in the manuscript appear to be odds ratios. Please confirm that prevalence ratios were calculated or correct this.
In cross-sectional studies, the prevalence ratio or proportion ratio, a measure used in this manuscript, is usually used, according to Schiaffino A et al. Odds ratio is used in cross-sectional studies. Gac Sanit 2003;17(1):70-4.

2
Abstract (lines 33-34): this statement (health outcomes…that underlie it) does not align with this study. No interaction was evaluated, no multi-level analysis was performed, and indeed no specific consideration of geographic scale was evaluated. This should be removed here and in the discussion (lines 431-432).
The lines suggested in the abstract and in the body of the manuscript are removed. Conclusions rewritten 

3
La The introduction is too long and insufficiently focused. The introduction should be a brief summary of the public health concern and specific considerations that builds to the study question. I would recommend radically shortening the introduction (perhaps by cutting lines 46-53, 67-70, and 90-100).

The introduction has been synthesized, following the recommendations of the evaluator.
4
No Sampling methodology is not sufficiently explained (lines 132-134): we need to know how many neighborhoods there were in the sampling frame (and how many ended up in the sample), how many blocks (median, minimum and maximum) in all neighborhoods (and then how many were selected in neighborhoods that ended up in the sample), and how many individuals were selected per block. Given the sampling methodology (line 133), the analysis likely should have used statistical weights to ensure that the survey responses were representative of the target population (older community dwelling adults in Medellin).
The information on the sampling carried out is expanded, in the Sampling section.
5
It is unclear what absolute and relative frequency measures are (line 142-143): relative to what? Response frequencies were calculated, overall and in strata of functional status.
The observation is followed and the frequency measurements used are specified, which for this case, are proportions.
6
What statistical measures were used in the estimation of physical performance (line 144-145)? Does this mean a regression analysis was done to determine physical performance? Or did the authors generate scores for the SPPB based on a previously-published method?
The wording of the statistical methods is adjusted to clarify that, from the score of the SPPB physical performance scale, two categories were generated: the first, including, "low and very low performance"; and the second, "moderate performance". Since no older adult achieved a score of 10 or more points on the scale, the high performance classification was not generated, so statistical analyses were made for this categorical variable in relation to the independent variables described through the estimation of prevalence ratios and confidence intervals (95%). 
Regression analysis was used to estimate crude and adjusted prevalence ratios.
7
Hypothesis tests aren’t calculated, they are performed (lines 146-147).
The oobservation is fallowed.
8
All bivariate associations reported in the tables are odds ratios, not prevalence ratios (line 150).
The prevalence ratio or proportion ratio (PR) has usually been used in cross-sectional studies as measures of non-causal association, given the limitations of these epidemiological designs; that is why in the manuscript the proportion ratio was used interchangeably as the prevalence ratio (PR).
9
No results are reported for the binary logistic regression (lines 150-151): This should either be removed from the manuscript or the results should be presented. In fact, it may benefit the authors to eliminate the unadjusted odds ratios presented in Tables 2-4 and instead present odds ratios adjusted for sex, age, marital status, housing type.
The observation is followed, and the measures adjusted for sex, age, marital status and type of housing are presented.

10
Language throughout the manuscript occasionally lapses into causal language (example line 152: “contribute to” should be “are associated with”; line 290: “influence this outcome” should be “are associated with”). This needs to be corrected throughout the manuscript.
The evaluator's suggestions were followed
11
Methods: a detailed description of the study variables and response options needs to be provided. The reader will be confused when they get to Table 1 and see familiar functionality/having been rejected by a health institution without any explanation of what is being reported.
The dependent variable and the independent variables are described in detail.
12
Table 1 (and 2-4): the presentation of frequencies and unadjusted odds ratios (no prevalence ratios are presented) in not necessary. The frequencies alone are sufficient (and the p-value for the chi-square test needs to indicate that this is for the test comparing moderate to low/very low functional autonomy- can be done in a footnote). I would recommend the authors replace the unadjusted ORs in Tables 2-4 and replace them with adjusted ORs from logistic regression models (see comment 9).
The observation is followed and the measures adjusted for sex, age, marital status, and type of housing are presented.

13
Tables 1-4: please keep only 2 decimal places for all measures of association (odds ratios and confidence intervals).
Observation is followed in all tables.
14
Results: the authors have repeatedly included parenthetical reference to “(RP=)”. I assume this is supposed to be PR=. Please correct throughout.
Corrected throughout manuscript (PR)
15
Results: the authors intermittently report survey respondent perception of the environment as if they were directly and objectively assessed by the researchers (e.g. line 220: in public parks there are not clean…). All statements of results based upon participant reported variables need to begin with “Participants reported…”.
The wording is adjusted in the results and in the methodology it is clarified that it is information of the participants
16
Discussion: Line 288- the statement that the elderly of Medellin have moderate physical performance can’t really be supported by this data if the analysis was not weighted to the sampling frame.
The suggestion is welcomed and the study participants are specified. 
17
Line 308: “what does none of them reached a satisfactory level” mean? Does this mean that their physical performance does not meet some fully functional status? This should be explained in the methods with regards to the scoring of the SPPB.  
Given the classification proposed by The National Institute on Aging.  Short Physical Performance Battery (SPPB) available in: http://www.nia.nih.gov/research/labs/leps/short-physical-performance-battery-spppb, .high (satisfactory) performance corresponds to having achieved at least 10 or more points on the SPPBB scale; however, none of the respondents obtained such a score. Although on the scale if they have moderate performance. It is explained in the methods within the item "Dependent variable".
To give greater clarity, in the line indicated by the reviewer it is corrected by "none of the older adults achieved a high performance (satisfactory); that is, it did not exceed nine points on the SPPB scale." 
18
Lines 309-330: This section of the discussion fails to acknowledge the potential for reverse causality. The direction of the association (self-owned homes leading to higher performance or higher performance leading to greater likelihood of living in self-owned home in old age).
The suggestion is acknowledged and declared as a bias within the limitations of the study
19
Line 354: “performance has a better…”, needs to be replaced with “performance reports a better…”.

The suggestion was followed; Change is made
"…reported higher physical performance when..."
20
Lines 374-375: “However,…existing communities” is not a full sentence. What is this supposed to be communicating?
The suggestion is welcomed. The sentence is deleted since it does not contribute to the discussion
21
Lines 390-396: Age discrimination is very abruptly introduced here. How does this apply: is it in planning for the needs of the elderly are subordinated to other groups?

The suggestion is accepted and clarified at the end of the paragraph that there are priorities with greater weight to age and that, therefore, make the elderly, a group that although not necessarily subordinate to other groups, do not constitute the population priority in terms of care and improvements for the achievement of their well-being.
22
Lines 405-415: This could use some more analysis- why is the situation in the Netherlands different from Colombia? Is it surprising that a high-income, densely populated, and not newly older (having had decades of experience with an aging population) country like the Netherlands may do a better job of meeting the needs of its older individuals?
The suggestion is accepted and a comparison of the built environments for Colombia and the differential conditions compared to the Netherlands is added. See second subchapter paragraph Built environment within the discussion
23
Line 431: “elderly is **associated with** physical social,…”.
The comment is welcomed and the conclusion is adjusted
24
Lines 432-434: This sentence doesn’t represent the value of this research (in fact, it overstates it). Exploring how environments are associated with performance may generate information to support design strategies to improve performance and maintain autonomy. The exploration of the environments does not in itself contribute to design strategies to improve performance and generate autonomy.
The comment is welcomed and the conclusion is adjusted

Reviewer 4 Report

The article is interesting, but two methodological issues must be corrected. First, the authors say:

An analytical, cross-sectional study was conducted that included 175 people over 60 years of age and their residential environments, whose homes were located in the urban area of ​​the city of Medellín Colombia and who agreed to participate; a probabilistic selection was used, with a bi-stage (neighborhoods and blocks) and random sampling

What is the level of confidence and the absolute error of the probabilistic sample -p=q=0.5-? This issue needs to be clarified. In any case, it seems that a sample of 175 cases is going to have a high absolute error, an issue that affects the significance levels of the PR. I think, therefore, that this limitation should be discussed in the text.

Second, the authors say:

Statistically significant association was considered with values ​​below 5% (p<0.05) and prevalence ratios (PR) with 95% confidence intervals (95% CI)

Wouldn't it be more correct to work with a p<0.01? The text has to talk about this statistical limitation.

Also, the authors say:

Our findings show that the population with a higher physical performance has a better social environment related to the space of their home, and attributes of the physical environment such as signaling, spaces for companions in public places, availability of suitable roads, sidewalks/separators, public transport routes, devices for speed control and for pedestrians… However, progress in planning these strategies to make age friendly cities in Latin American countries and Colombia remains a challenge

It seems important to clarify what specific aspects of some urban policy would have to change in Medellín/Colombia based on the empirical evidence of the article. The reflections shown in the article are very general.

Author Response

The authors of the referenced manuscript, appreciate your comments and recommendations for the improvement of a version more adjusted to those expected by the International Journal of Environmental Research and Public Health and therefore we respond to each of them:
Comments and Suggestions
Answers
The article is interesting, but two methodological issues must be corrected. First, the authors say:
An analytical, cross-sectional study was conducted that included 175 people over 60 years of age and their residential environments, whose homes were located in the urban area of ​​the city of Medellín Colombia and who agreed to participate; a probabilistic selection was used, with a bi-stage (neighborhoods and blocks) and random sampling
What is the level of confidence and the absolute error of the probabilistic sample -p=q=0.5-? This issue needs to be clarified. In any case, it seems that a sample of 175 cases is going to have a high absolute error, an issue that affects the significance levels of the PR. I think, therefore, that this limitation should be discussed in the text.
A sampling section was added to methods and material, where the requested information was added
The authors say:
Statistically significant association was considered with values ​​below 5% (p<0.05) and prevalence ratios (PR) with 95% confidence intervals (95% CI)
Wouldn't it be more correct to work with a p<0.01? The text has to talk about this statistical limitation.
The sampling explains that the confidence level used was 95%, which is why it is not possible to modify the level of statistical significance.
The authors say:
Our findings show that the population with a higher physical performance has a better social environment related to the space of their home, and attributes of the physical environment such as signaling, spaces for companions in public places, availability of suitable roads, sidewalks/separators, public transport routes, devices for speed control and for pedestrians… However, progress in planning these strategies to make age friendly cities in Latin American countries and Colombia remains a challenge
It seems important to clarify what specific aspects of some urban policy would have to change in Medellín/Colombia based on the empirical evidence of the article. The reflections shown in the article are very general.
The suggestion is welcomed and after this paragraph, the context of Medellín's urban policy in the face of aging and old age in terms of the environment is clarified, in order to explain why the limited progress in improving these attributes continues to be a challenge in the local and national context.

Round 2

Reviewer 2 Report

Thanks for addressing my comments, I think the manuscript has greatly improved. I have another comment regarding the data analysis: why did you not performed a multivariate regression analysis? This would allow for ascertaining the main determinants of functional autonomy. 

As minor comments, please, correct the commas in the decimals in table 2. Also, in the footnotes in Tables 2 and 3 there are asterisks that do not correspond to data in the tables.

Author Response

I attach the answer to your comments. Thank you

Reviewer 3 Report

Overall Comments:

This study reports on the associations between perceived environmental domains and functional autonomy among older residents of Medellin, Colombia. The revised manuscript is greatly improved, but requires further attention to the introduction, methods, and results.

Comments:

1.      Lines 34-36: this is a sentence fragment, and doesn’t follow from the abstract. I would cut these, as they don’t add to the conclusions.

2.      Line 42: Replacing “develop” with “maintain” would be more accurate and follow from the language in the abstract (line 34- maintain autonomy).

3.      Line 45: Adding “greater” before “well-being” would add clarity.

4.      Line 47: unclear who the subject of the sentence is. Who has considered the design of age friendly cities to promote healthy lifestyles? This needs to be specified.

5.      Line 51: Paragraph begins with an unclear subject. Who is represented by “Its”? This needs to be specified.

6.      Line 57: adequation is not a word in English. Adequacy is the appropriate word.

7.      Lines 59-60: opening sentence of this paragraph is not appropriate for an academic work- this is a directive political statement. The content can either be removed or rephrased (remove “must”).

8.      Line 77: Autonomy “of” not autonomy “from”.

9.      Line 104: What is meant by the sampling error? How was this calculated? A citation for what this number means for the results of the study would be helpful.

10.   Line 127: APGAR needs to be replaced with “APGAR Familiar” (and I would italicize it too). For English-speaking audiences, APGAR is a distinct score for the assessment of infants after birth.

11.   Lines 132-139: Please specify that the comparison being made is between low/very low functioning and moderate functioning somewhere in this section. Please also indicate what covariates are adjusted for in each regression model.

12.   Line 138: Table 1 claims to present crude prevalence ratios, but all measures presented are in fact crude odds ratios. I am unable to determine whether the remaining tables present odds ratios or prevalence ratios, as those results are adjusted for covariates and cannot be calculated from the frequencies in the table.

13.   Line 156-158: “contribute the most” is causal language. This should be rephrased to “are the most strongly associated with”. In fact, you describe a number of associations, but none are statistically significant- so nothing is associated with your outcome. Please remove the sentence from lines 156-158.

14.   Line 162: As indicated in comment 12, this table presents crude odds ratios, not prevalence ratios. Please correct either the label or the measure of association, and confirm that the results in Tables 2 (line 196), 3 (line 228) and 4 (line 267) are correctly labeled (i.e. are these adjusted prevalence ratios, or adjusted odds ratios).

15.   Tables 1-4: Measures of association should be rounded to 2 decimal places.

16.   Tables 2-3: These tables have footnotes for the model R2. Does this mean that all of the variables presented in the table were included in a single regression model? Please specify in the methods.

17.   Table 4: Need a footnote for the regression model to indicate model type and whether these measures of association were adjusted for anything.

18.   Table 4: First variable (first 2 rows) is missing the variable name.

19.   Line 167: replace “old age” with “older residents”.

20.   Line 178: “not have adequate urban facilities” is confusing. Does this mean there are “not adequate urban parks”?

21.   Lines 179-180: It isn’t clear why the sentence about commercial/religious establishments follows from the prior sentence about parks. I would cut this sentence.

22.   Lines 235-263: Focus on interpreting only significant measures of association. If you don’t have any statistically significant measures of association, state that.

23.   Line 258: public transport is not associated with functional autonomy (the confidence interval contains 1.00).

24.   Line 259 (“increases”) and line 263 (“limited”): this is causal language for the associations, indicating both direction of association (which can’t be known in a cross-sectional sample) and that the association is causal.

25.   Lines 341-343: Are all of these variables statistically significant. It is unclear to me, because the labeling is not consistent between this sentence and Table 4 variable labels.  

26.   Line 463: instead of “axes” I would use “principles”

27.   Line 468: “age at an accelerated pace” does not make sense. Why would we expect the rate of aging to be higher in the future?

Author Response

(The authors gave the same response as above.)

Reviewer 4 Report

Congratulations for your paper.

Author Response

(The authors gave the same response as above.)
